# Polymeric Coatings for AR-Glass Fibers in Cement-Based Matrices: Effect of Nanoclay on the Fiber-Matrix Interaction

**Francesca Bompadre** [1,*] , **Christina Scheffler** [2] , **Toni Utech** [2] and **Jacopo Donnini** [1]

1   Department of Materials, Environmental Sciences and Urban Planning, Università Politecnica delle Marche, 60131 Ancona, Italy; jacopo.donnini@staff.univpm.it
2   Leibniz-Institut für Polymerforschung Dresden, Hohe Strasse 6, 01069 Dresden, Germany; scheffler@ipfdd.de (C.S.); utech@ipfdd.de (T.U.)
*   Correspondence: f.bompadre@pm.univpm.it

**Abstract:** Polymeric coatings are widely used to enhance the load bearing capacity and chemical durability of alkali-resistant glass (AR-glass) textile in cement-based composites. The contact zone between coated yarns and concrete matrix plays a major role to enable the stress transfer and has still to be improved for the full exploitation of the mechanical behavior of the composite. As a new approach, this paper studies how the addition of nanoclay particles in the polymer coating formulation can increase the chemical bond between organic coating and inorganic matrix. This includes the description of the water-based coating preparation by dispersing sodium montmorillonites, whereby the resulting coating nanostructure is characterized by X-ray diffraction and energy dispersive X-ray spectroscopy. Single glass fibers were treated by dip-coating. Atomic force microscopy was used to determine the surface roughness, and the effect on the fiber tensile properties was studied. Moreover, the morphological and chemical characteristics of the coatings were compared with the results obtained from single fiber pull-out (SFPO) tests. It was shown that the incorporation of nanoclays leads to increased interfacial shear strength arising from the ability of nanoclay particles to nucleate hydration products in the fiber-matrix contact zone.

**Keywords:** alkali-resistant glass fiber; polymer coating; nanoclay; interface; cement-based matrix; glass fiber reinforced concrete composites





## 1. Introduction

Since the first employment of asbestos cement in the 1900, the use of synthetic fibers for the reinforcement of existing constructions has become a well established practice which finds different usages in civil engineering applications [1]. Particularly, continuous multi-filaments textile embedded within inorganic based matrix proved to be particularly effective to strengthen and rehabilitate concrete and masonry structures [2–5]. These systems are known as Fabric Reinforcement Cementitious Matrix (FRCM) or Textile Reinforced Mortar (TRM) and present several advantages compared to traditional retrofitting techniques, such as high strength to weight ratio, corrosion resistance, easy and fast application, and low invasiveness [2–6]. Although fabrics are usually constituted by yarns made of dry filaments, different studies and commercial solutions propose the employment of fabric impregnated with polymeric coatings, usually epoxy resin or styrene-butadiene copolymer [7–10]. The application of the coating has the double effect of enhancing the load bearing capacity of the reinforcement [8,11–14] and protecting the fibers from abrasion and chemical corrosion [9,15–19], which is of particular significance for alkali-sensitive fibers, such as glass fibers. The increased mechanical performances of FRCM systems with coated fabrics is mainly attributed to the improved stress transfer capacity between the filaments [10–13], which, in dry FRCM systems, is compromised by the poor penetration of the inorganic matrix within the filaments [11]. When a polymeric coating is applied, the load transfer mechanism between fabric and matrix changes [20]. In dry systems, the tensile behavior of

the composite depends on the bond between filaments and matrix, while the bond between neighboring filaments is almost negligible [4,11]. On the contrary, in pre-impregnated yarns, the fibers are embedded in the coating, which prevents the matrix to penetrate into the yarns. As a consequence, the stress-transfer between the fabric and matrix no longer depends on the ability of the latter to penetrate between the individual filaments, but on the interaction of the coating with the cementitous matrix. Experimental studies showed that, by applying a polymeric coating, the failure mechanism of the composite shifts from slippage of the filaments inside the yarn, to slippage of the yarns into the matrix and delamination at the fabric-matrix interface [8,10]. This is due to difference in the chemical characteristics of organic coatings (hydrophobic) and inorganic matrix (hydrophilic), which can lead to a relatively poor bond of polymers with the cementitious matrices. In order to enhance the chemical interaction between these two components, some studies propose the application of inorganic coatings [21–25]. However, polymeric coatings present some characteristics that make them particularly suitable for industrial production. First of all, surface modification of fibers by application of sizings and coatings is a standard procedure, and industrial techniques for the production of coated fabrics are already on the market [26,27]. Secondly, because of the great variety of polymeric coatings available, the specific characteristics of the fabric can be adjusted, for example, by providing a good flexibility which facilitate the application on the substrate. A promising, but still relatively unexplored technique to maintain the versatility of organic coatings for textile reinforcement, while enhancing their chemical compatibility with inorganic matrices, is the incorporation of inorganic nanofillers in the polymer [28–30]. Inorganic nanofiller, typically employed for the production of polymer nano-composite, includes: carbon nanoparticles, carbides, nanoclays, and nano-oxide, particularly silica ($SiO_2$), alumina ($Al_2O_3$), and titania ($TiO_2$) [31]. Polymer nano-composites generally exhibit better properties compared to the plain polymers, such as better mechanical properties, thermal and dimensional stability, fire and chemical resistance, gas permeability, and optical and electrical properties [32–35]. The superior reinforcing effect obtained with the inclusion of nanoparticle in the polymer matrix compared with that of microscale particles is due to the large nanoparticles' interfacial area. This has a significant impact on the final properties of the polymer nanocomposite, since the polymer at the interface present different properties than the bulk polymer, such as a higher degree of crystallinity [36]. Among the possible nanofillers that can be used for this purpose, nanoclay particles are particularly attractive. Clay is a relatively cheap and easily available material consisting of layered silicate that finds several industrial applications, ranging from cosmetics and food industry to production of high performance materials [35,37–39]. Clays are a mineral belonging to the structural family of phyllosilicates and are referred to as layered silicates due to their lattice structure consisting of octahedral sheet of alumina or magnesia fused to one or two external silica tetrahedra forming thin layers that bound together with counter-ions. In the construction field metakaolin, a calcined form of clay with mean particle size of about 3 μm, has been widely employed as pozzonalic binder in mortar and concrete matrix. The incorporation of metakaolin as a partial cement replacement has been reported to influence the properties of the fresh paste, as well as the mechanical and durability performance of the cured matrix [40]. Particularly, cement and mortars containing a proper amount of metakaolin exhibit higher compressive and flexural strength, as well as increased durability and reduced shrinkage, compared to the same matrix without metakaolin [40]. Matrices containing metakaolin have also been employed for TRC composites, which benefit from the small dimension of the binder, its pozzolanic properties, and its ability to inhibit alkali-silica reactions [41,42]. Although addition of nanoclay particles in concrete is less common, promising results have been obtained in terms of mechanical properties and durability [43]. Particularly, clay dispersed in concrete matrix has been reported to act as nucleation agents for the formation of Calcium Silicate Hydrate (CSH) [44,45]. Moreover, it was previously shown that the modification of textile reinforcement with a polymeric coating containing nanoclay particles can promote formation of hydration products at the

fiber-matrix interface [29,30]. Another property of nanoclay polymers that make them particularly suitable for modification of reinforcements used in TRC composites is the ability of nanoclay polymers to act as moisture barrier. This is attributed to the tortuous path that permeating molecules have to traverse moving between the nanoclay plates dispersed in the polymer [35]. When the nanoclay polymer is applied on the surface of glass fibers, it forms a barrier layer which improves their resistance to alkaline corrosion (Figure 1) [46].

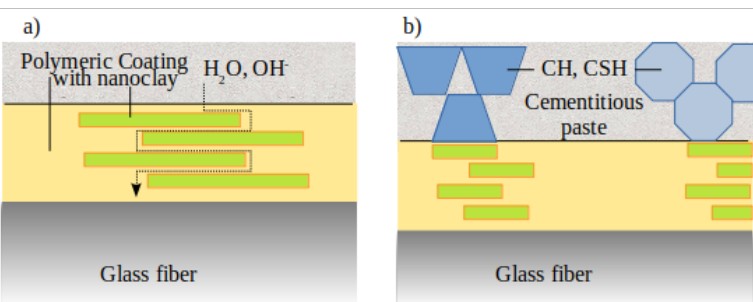

**Figure 1.** (**a**) Structure of a silicate layered coating on the surface of glass fibers and its barrier properties according to Gao et al. [46]; (**b**) nanoclay particles dispersed in the coating promoting the formation of hydration products at the fiber-matrix interface.

However, it should be stressed out that a prerequisite for an effective improvement of the polymer barrier properties is the proper dispersion of the nanoclay in the polymeric matrix. Indeed, any physical mixture of polymer and nanoclays does not necessarily results in the formation of a nanocomposite [35]. This is obtained only if the nanoclay in the polymer presents either an intercaleted or exfoliated structure. In the first case, the nanoclay polymer has a multi-layers structure of alternating polymeric and silicate layers. In the second case, the clay layers are completely separated from each other and individually dispersed in the polymeric matrix. According to the chemical physical characteristics of the polymer employed, different techniques can be used for the preparation of a polymer-clay nanocomposite. These include: exfoliation-adsorption, in situ intercalative polymerization, melt intercalation, and template synthesis [47]. The incorporation of layered nanoparticles at the interphase of fiber reinforced composites is also considered a promising strategy to increase the composite's toughness [48–51]. Encouraging results have been obtained for fiber reinforced polymer/clay nanocomposite [48,51,52]. The ability of nanoclay to increase the fracture toughness is attributed to a combination of mechanisms which include: deflection and pinning of the crack propagation, prevention of the crack formation, and stop crack propagation due to mechanical interlocking and void bridging [53]. The ability of nanoparticles to improve the mechanical properties of fiber reinforced composites has also been supported by computational studies [49,50,53,54], that highlighted that the nanoplatelets localized in the fiber/matrix interfacial layer are more effective at crack deflection than nanoplatelets dispersed throughout the matrix [49,54]. However, the possibility to modify the fiber-matrix interface of cementitious based composites with coating containing nanoclay particles has been poorly investigated. Previous studies have shown that the nanoclay particles can promote the formation of hydration products at the fiber matrix interface [29,30]. Nonetheless, to the authors' best knowledge, the contribution of nanoclay particles to the fiber-matrix interfacial bond has never been characterized.

In this work, sodium montmorillonite (MMT) nanoparticles were added to a styrene-butadiene (SB) coating and applied to AR-glass fibers produced by a lab scale spinning device without surface treatment during fiber manufacturing. Energy-dispersive X-ray spectroscopy (EDX) and X-ray diffraction (XRD) were employed to investigate the dispersion of the MMT nanoparticles in the polymeric matrix. Single fiber tensile tests were conducted on unsized fibers, as well as sized and coated fibers. The effects on the fiber-matrix interaction were investigated by means of micro-mechanical single fiber pull-out (SFPO) tests. The pull-out behavior of fibers modified with different coatings, with and

without addition of nanoclay, was compared with that of fibers treated with only organosilane coupling agent. Atomic Force Microscopy (AFM) and Scanning Electron Microscopy (SEM) analyses were conducted to correlate the mechanical results obtained with the morphological and chemical properties of the fibers surface. Growth of crystals on the fibers surface was investigated with a digital microscope and employed as qualitative method to measure the effect of nucleation on the formation of hydration products.

## 2. Materials and Methods

### 2.1. Fibers

Alkali-resistant glass (AR-glass) fibers, with diameter of $13.4 \pm 1.3$ µm, were made at the Leibniz Institut für Polymerforschung Dresden, Germany (IPF), by using a continuous pilot plant spinning equipment. The application of a sizing during the glass spinning process is crucial for subsequent usability and further processing. However, it was found in previous experiments that single fibers isolated from spun-sized strands present regions where sizing is accumulated between filaments and form a non-homogeneous layer on the surface. Therefore, in micro-mechanical experiments, the dip-coating of each single fiber is a more suitable procedure, since it provides a significantly more homogeneous sizing or coating layer compared to fibers that were surface treated in the spinning. On the other hand, it must be mentioned that the high effort to dip-coat each single fiber is limited to such a fundamental work as provided in this paper. For dip-coating, one filament was separated from a strand of unsized AR-glass filaments. Then, one fiber end was fixed to a small plate and the free end was dipped into the silane solution using a constant velocity. Further modification of the fiber surface with additional polymeric coatings was conducted with the same method followed by a fiber drying at 140 °C over 4 h in an oven, as schematized in Figure 2. Table 1 reports the composition of the sizings and the coatings used. A water solution containing 1% of 3-Aminopropyltriethoxysilane (AMEO) was used in the 1st treatment step for the coated fibers. However, the use of AMEO was found to be unsuitable for the scale up of the sizing process, since the amino moiety of the silane reacts with the carboxylated styrene butadiene used as film former in the sizing formulation. For this reason, the fibers used to investigate the interaction of the matrix with the sized fibers were modified with N-Propyltrimethoxysilane (PTMO) instead of AMEO. Both N-Propyltrimethoxysilane and 3-Aminopropyltriethoxysilane were purchased from EVONIK industries (Essen, Germany).

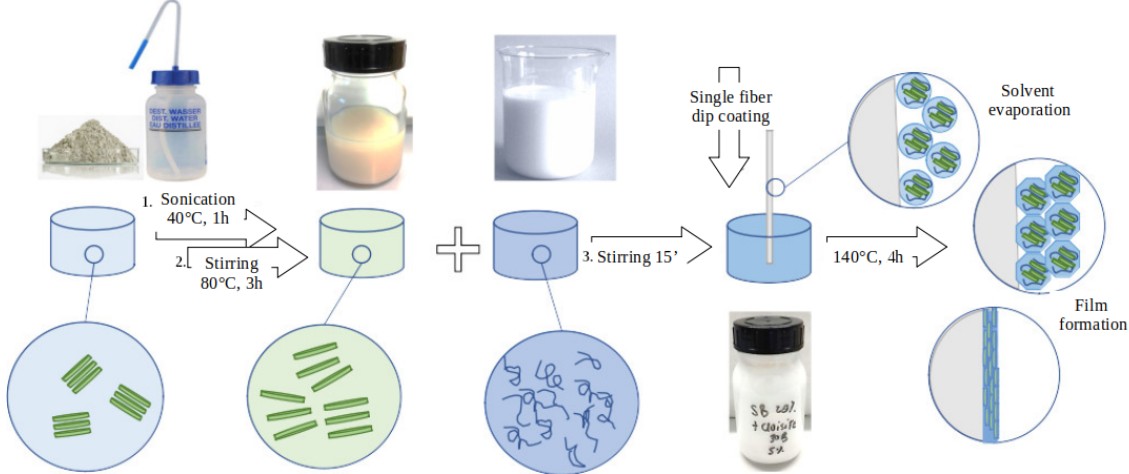

**Figure 2.** Schematic representation of dispersion preparation and fiber surface treatment process.

Images of the fibers before and after modification were acquired with a Ultra plus (Carl Zeiss NTS, Germany) microscope in high vacuum and an accelerating voltage of 3 kV.

Information about the morphology of the coated fibers were obtained by Atomic force microscopy conducted with a Dimension Icon (Bruker Corporation, Billerica, MA,

USA) equipped with a Tap300-G BudgetSensors (Innovative Solutions Bulgaria Ltd., Sofia, Bulgaria) cantilever made of monolithic silicon with a rotated pyramid shaped tip and a tip radius less than 10 nm. Roughness parameter in terms of arithmetic average roughness ($R_a$), square roughness ($R_q$), and maximum roughness ($R_{max}$) were obtained recording topography images with dimension 3 μm × 3 μm and 512 px × 512 px in tapping mode. Results were reported as the average of at least 3 images for each fiber type.

**Table 1.** Surface modification of ARG fibers.

| Sample Name | Sizing (1st Treatment Step) | Coating (2nd Treatment Step) |
|---|---|---|
| PTMO | 1% N-propyltrimethoxysilane (PTMO) in distilled water | none |
| SB | 1% 3-Aminopropyltriethoxysilane (AMEO) in distilled water | self-crosslinking styrene-butadiene copolymer (SB) |
| CSB | 1% AMEO in distilled water | SB + 7% crosslinking agent (C) |
| SB-MMT | 1% AMEO in distilled water | SB + 5% montmorillonite (MMT) |
| CSB-MMT | 1% AMEO in distilled water | CSB + 5% MMT |

Effect of the sizing and coating on the fiber tensile strength was investigated by means of single fiber tensile test performed with a Favigraph testing device (Texchno, Mönchengladbach, Germany) equipped with a 1 N load cell. The fiber diameter of each filament was determined from the linear density obtained with an integrated vibroscope and considering a density of glass fibers of 2.71 g/cm$^3$. The tensile strength was calculated as the fracture load divided by the fiber cross sectional area. The testing velocity was 10 mm/min at a gauge length of 20 mm.

## 2.2. Coating Preparation and Characterization

Sodium montmorillonite (MMT), with a typical dry particle size < 25 μm and basal spacing of 1.17 nm ($d_{001}$ value), was obtained from Southern Clay Products, Inc., under the trade name of Cloisite®Na$^+$. A carboxylated and self-crosslinking styrene-butadiene latex (SB), with a solid content of 50%, was supplied by Lefatex Chemie GmbH (Brüggen-Bracht, Germany) along with the methylol melamin crosslinking agent (C, solid content 75%). Preliminary tests showed that no stable dispersion could be obtained by directly mixing the nano-clay in the polymeric latex. For this reason, a different protocol was developed in which the nano-clay particles were first dispersed in distilled water and successively mixed with the SB latex (Figure 2):

1. First, 2.5 g of nano-clay particles were slowly added to 50 mL distilled water in a 100 mL round flask under vigorous stirring. The mixture was stirred for about 5 min and then was placed in an ultrasonic bath at 40 °C for 1 h.
2. The mixture was then refluxed for 3 h at 80 °C until a homogeneous dispersion was obtained. Once the dispersion was cooled down, it was filtered with a 14–19 μm filter in order to remove possible agglomerated particles. In order to exclude the presence of agglomerates, the particles size distribution of the nanoclay in the aqueous dispersion was analyzed with Malvern Zetasizer Nano ZS Particle and ZetaPotential Analyzer. The distribution obtained was found equal to that reported in the technical sheet for Cloisite®Na$^+$, which indicates that the nanoclays were well dispersed;
3. The nanoclay dispersion was stirred for 15 min with the SB latex and distilled water. The proportion between nanoclay dispersion (MMTD), SB latex (SB), and distilled water ($H_2O$) was calculated as follows: first, the amount of SB latex and MMTD dispersion were determined according to Equations (1) and (2), and then the obtained dispersion was diluted with distilled water in order to obtain a final solid content of 20% (Equation (3)).

$$m_{SB}[g] = \frac{x \cdot m_{SBMMT_x}[g]}{scSBC[\%]} \cdot 100$$

$$with \quad x = \frac{0.2}{1.05}$$

(1)

$$m_{MTTD}[g] = \frac{x \cdot m_{SBMMT_x}[g] \cdot 0.05}{scMMTD[\%]} \cdot 100,$$

(2)

$$m_{H_2O} = m_{SBMMT_x}[g] - m_{SB}[g] - m_{MTTD}[g],$$

(3)

where:

$x$ is the polymer solid fraction in the SB-MMT coating;
$SBMMT_x$ is the desired amount of SB-MMT coating expressed in gram;
$scSBC$ is the SB latex solid content determined according to DIN EN ISO 3251;
$scMMTD$ is the solid content of the MMTD dispersion determined according to DIN EN ISO 3251.

Coatings CSB and CSB-MMT were obtained adding the methylol melamin cross-linking agent respectively to the SB and the SB-MMT dispersion under stirring.

The characterization of the coatings was conducted on the polymer films, obtained by placing some grams of the nanoclay-latex dispersion in a Teflon Petri dish and letting it curing in oven at 140 °C for 4 h. For the investigation of the coating nanostructure, XRD analyses were conducted with a Bruker D8 Advance difractometer, with incident wave length radiation of 1.54060 Å. Data acquisition was performed between 2° and 30° using a scanning speed of 2 s/step.

The results obtained were coupled with SEM-EDX images of the coatings SB-MMT and CSB-MMT in order to obtain further information about the dispersion grade of the nanoclay in the polymeric matrix. For this purpose, a SEM Ultra Plus (Zeiss, Oberkochen, Germany) with a EDX-Detector X-Flash 5060F (Bruker, Berlin, Germany) was used.

### 2.3. Cementitious Matrix

The cementitious matrix was designed according to the typical composition of pre-mixed commercial products, and it is constituted of a mix of cement, hydrated lime, calcium carbonate, and a redispersible dry polymer vinyl acetate/vinyl esther of versatic acid/ethylene (VAc/VeVa/E), in the following proportions:

- CEM II/B-LL 42.5 R—165 kg/m$^3$;
- CEM II/B-LL 32.5 R—83 kg/m$^3$;
- Hydrated lime—110 kg/m$^3$;
- Calcium carbonate 600—206 kg/m$^3$;
- Calcium carbonate 400—715 kg/m$^3$;
- Vac/VeVa/E—21 kg/m$^3$.

Before mixing with tap water, the matrix was ground with a mortar to facilitate the realization of the single fiber composite samples for pull-out. Further information about the mechanical properties of the matrix can be found in Reference [55].

### 2.4. Single Fiber Pull-Out Test

Fibers were embedded in the cementitious matrix using a special equipment designed and constructed at IPF [56–58]. For the realization of the pull-out specimens, the fine grained matrix was mixed with tap water with a speed mixer. The same matrix has been employed in previous study with a water/binder ratio of 0.7 [55]. However, in this work, a water/binder ratio of 1.14 was used. This was necessary in order to obtain a fluid paste that enables the realization of matrix droplets with an adequate shape. The mixture was then filled in a sample holder with cylindrical shape and an inner diameter of about 2.6 mm. One end of the fibers was embedded by a computer-controlled process for 1000 μm length under controlled atmosphere and temperature (23 °C, 50% relative humidity (RH)).

About 19–20 specimens were prepared for each coating and cured for 28 days at constant temperature of 23 °C and 90% RH. The SFPO tests were conducted using a specialized pull-out apparatus, for which a detailed description can be found in Reference [56,58]. The single fiber model composite was placed in the pull-out device and the upper fiber end was fixed at a mandrel with a cyanoacrylate adhesive in such a way that the free length was minimized. Subsequently, the pull-out test was performed under quasi-static loading conditions using a loading rate of 0.01 μm/s.

### 2.5. Hydration Products

The addition of nanoclay particles in polymeric coatings was shown to facilitate the formation of hydration products at the fibers-matrix interface [29,30]. A deep understanding in the formation of hydration products on the polymer surface is of primary importance to evaluate the effects of nanoclay. However, when fibers are extracted from the cementitious matrix, the fiber-matrix interface is damaged, so that it is difficult to obtain information about the accretion of hydration products. For this reason, an alternative method was used, in which the fibers SB and SB-MMT were both immersed in an aqueous concrete solution. The solution was obtained by adding 40 g of cementitious matrix in 200 mL distilled water in a volumetric flask. The flask was ultra-sonicated for 1 h, after which the solution was filtrated with a 10 to 15 μm filter. The fibers were immersed in the solution and let stay overnight. Then, they were removed from the solution, washed for 1 h in distilled water, and successively observed with a Digital-Microscope Keyence VHX2000 to investigate the possible formation of hydration products.

### 3. Results and Discussion

### 3.1. Coatings Nanostructure

X-ray diffraction has been widely employed for the characterization of polymer-layered silicate (PLS) nanocomposites [35]. In fact, the gap between the silicate layers (*d*), called "interlayer" or "gallery", can be determined according to the Bragg's law:

$$sin\theta = \frac{n\lambda}{2d}, \tag{4}$$

where *θ* is the diffraction angle measured, and *λ* corresponds to the wave length of the X-ray radiation used in the diffraction experiment [59]. Therefore, the X-ray diffractogram of the nanoclay polymer can be used to determinate if an intercalated or exfoliated nanostructure has been obtained. In the first case, the polymeric chains must intercalate between the silicate platelets causing an increase of the interlayer spacing. Therefore, the diffraction peaks obtained for an intercalated nanoclay polymer shifts towards lower angle compared to the XRD patterns of the pure nanoclay powder. In case of an exfoliated PLS nanocomposite, the ordered structure of the nanoclay is lost, so that no more diffraction peaks are visible in the XRD diffractograms. Figure 3 shows the diffractograms of the as received Cloisite®Na$^+$ powder, a nanoclay film obtained by drying the aqueous nanoclay disperion (MMTD) at 110 °C (Figure 3b), and polymeric films of the SB-MMT (Figure 3c), as well as CSB-MMT (Figure 3d).

The layers spacing calculated from the diffraction peak of the as received Cloisite®Na$^+$ is equal to 1.21 nm, which is in accordance with the information reported from the producer. It can be observed that the diffraction peak of the nanoclay film obtained form the MMTD is shifted towards lower angle compared to the XRD patterns of the as delivered Cloisite®Na$^+$ powder. This suggests that the distance between the layered silicates has been increased. Montmorillonite, as other smectite clay, is reported to adsorb water molecules and form stable aqueous dispersion where the interlayer spacing can increase up to 4 nm for high water content [60]. By successively drying the dispersion, the silicate interlayer spacing falls back to smaller distance value. However, Na$^+$ cations in the silicate gallery are reported to form hydrated complex with strongly bonded water molecules [61]. Therefore, it can be supposed that the shift of the diffraction angle observed for the Cloisite®Na$^+$ after dis-

persion in distilled water is due to the hydration of the montmorillonite interlayer cations. Figure 3 shows that the diffraction peaks of the SB-MMT and CSB-MMT film are further shifted to smaller angles, with a corresponding *d* value of 1.46 nm. The diffraction angles observed for sample SB-MMT and CSB-MMT indicate a greater separation of the silicate layers. However, the obtained space gallery is comparable with the intercalation of small molecules, such as alkylammonium ions or organosilane molecules [62,63], rather than polymeric chains [64,65]. Since surfactants are commonly employed for the production of polymeric latex, the observed increase in the diffraction angles can also be attributed to the intercalation of organic ions between the silicate layers. This mechanism would reassemble the intercalation of alkylammonium ions typically employed for the functionalization of layerde silicates with organic moieties [35]. Therefore, according to the results obtained, the formation of a nanostructured polymer cannot be confirmed. The flatter shape of the CSB-MMT diffraction peak (Figure 3d) can be attributed to a lower signal intensity due to the formation of many air bubbles in the polymeric film, as can be observed from Figure 4c.

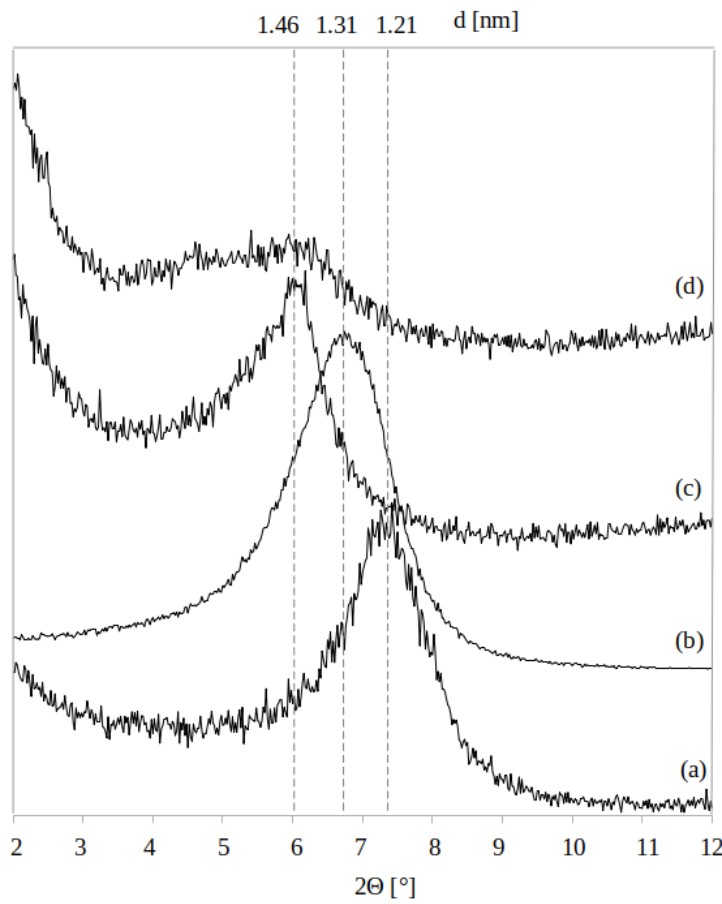

**Figure 3.** XRD diffractograms of (**a**) MMT powder, (**b**) film obtained from the MMTD, (**c**) SB-MMT film, (**d**) CSB-MMT film.

Although XRD offers a conventional method for the determination of the nanocomposite interlayer structure, it does not give any information about the spatial distribution of the silicate in the matrix or the formation of inhomogeneties in the PLS nanocomposite. In order to obtain further information on the dispersion of the MMT in the SB coating, XRD results were coupled with SEM-EDX analyses. These were carried out on the same polymeric films used in the XRD. SEM-EDX images gave different results for sample SB-MMT and CSB-MMT. In both cases, the elemental analyses confirm the presence of atoms typical for clay composition, such as Mg, Al, and Si. However, Figure 5 shows that these elements do not present the same distribution in the two polymeric matrices. As it can be observed

from the elemental distribution of Si, the nanoclay in SB forms a kind of oriented strips of about 100 µm length and 5 to 50 µm width uniformly dispersed in the matrix. On the contrary, CSB-MMT forms agglomerates of more than 100 µm width, indicating that the nanoclay presents a worse distribution in the CSB polymer than in the SB.

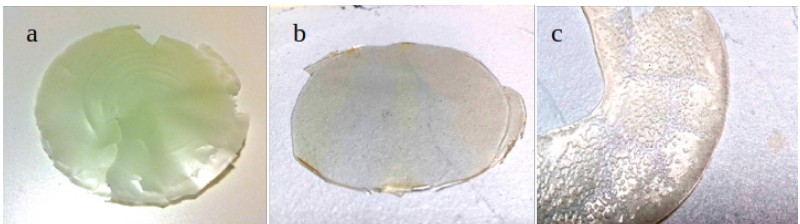

**Figure 4.** Film of (**a**) MMT, (**b**) SB-MMT, (**c**) CSB-MMT used for RXD and SEM analyses.

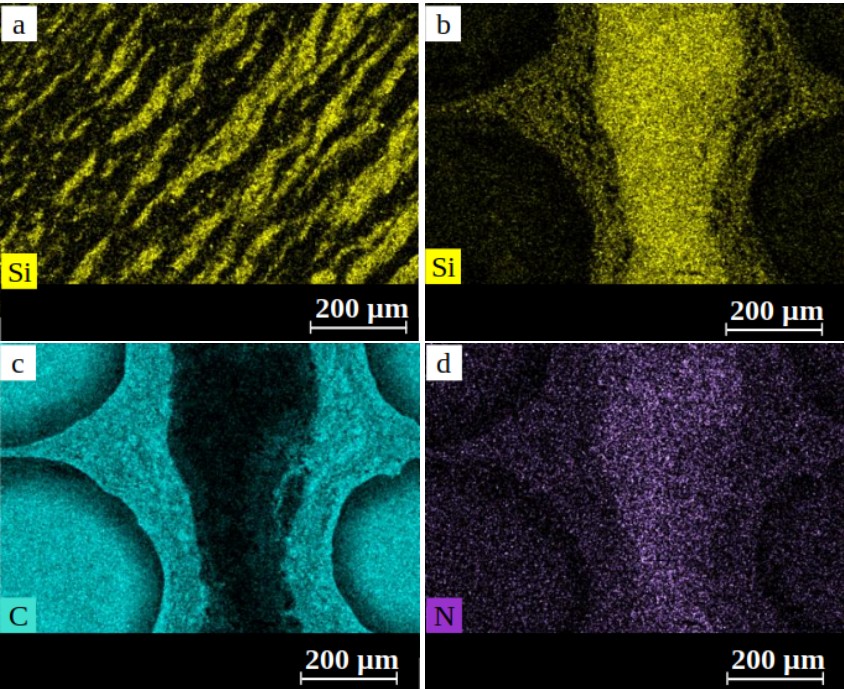

**Figure 5.** Elements distribution in polymer films of SB-MMT (**a**) and CSB-MMT (**b–d**). Magnifications 100×.

The main element detected in both polymers is carbon, which is the principal component of the SB coating. In the CSB-MMT sample, nitrogen atoms could also be detected, which can be attribute to the methylol melamine molecules of the cross-linking agent. It can be observed in Figure 5 that the distribution of C and N atoms does not perfectly match. N seems to be more concentrated in the nanoclay agglomerate, rather than homogeneously dispersed in the polymeric matrix. This suggests a correlation between the formation of nanoclay agglomerates and the addition of the cross-linking agent. This can be due to the addition of a new chemical component in the dispersion, which increases the number of possible interactions that can occur between polymer and nanoclay, influencing the development of the coating structure [66]. Moreover, the marked difference in the distribution of the C and N atoms in the CSB-MMT matrix indicates that the reaction between the carboxylated styren-butadiene chains and the methylol melamine cross-linking agents did not uniformly occur. To the authors' best knowledge, no studies are available about the effect of nanoclay on the cross-linking process of carboxylated styrene-butadien with methylol melamin. However, some studies have reported that nanoclays can influence the curing process of polymers [67,68]. This is attributed, on one hand, to the reduction of the molecular mobility, caused by an increase in the viscosity of the polymer dispersion,

because of the incorporation of nanoclay particles [69]. On the other hand, nanoclays can also influence the reaction mechanism that governs the curing process [68]. Therefore, it can be supposed that, in presence of nanoclay, the methylol melamin cross-linking agent has not fully reacted with the carboxylated styrene-butadiene copolymer, but, rather, it has concentrated around the nanoclay agglomerates.

### 3.2. Fiber Surface Morphology

The presence of agglomerates in the CSB-MMT polymer results in the formation of a non-homogeneous coating on the fiber surface, as can be observed in the SEM images reported in Figure 6. Although all the fibers were coated with the same polymeric dispersion, using the same procedure, it can be seen that they deeply differ one from the other.

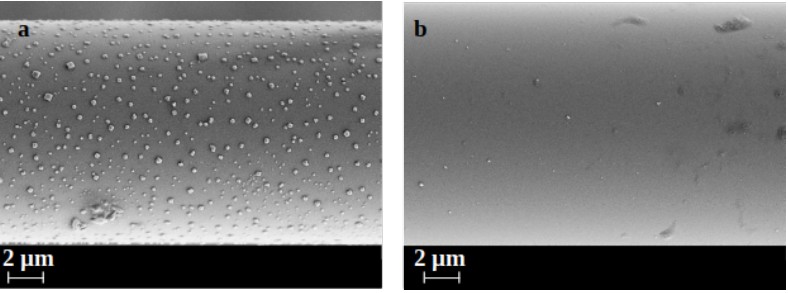

**Figure 6.** SEM images of two fibers coated with CSB-MMT. Magification 2.00 K×.

The fiber in Figure 6a presents on its surface some agglomerates of dimensions varying from some tens of nanometers to almost one micrometer. The agglomerates differ for dimension and distribution on the fibers surface. These agglomerates cannot be seen on the other fiber (Figure 6b), which shows a much more homogeneous surface. On the contrary, Figure 7 shows that all the other coatings form a homogeneous film on the fiber surface.

Figure 8 displays AFM surface topography imaging and mean roughness values of the fibers modified with the four different coatings previously described. The average roughness $R_a$, square roughness $R_q$, and maximum roughness $R_{max}$ are reported in Table 2.

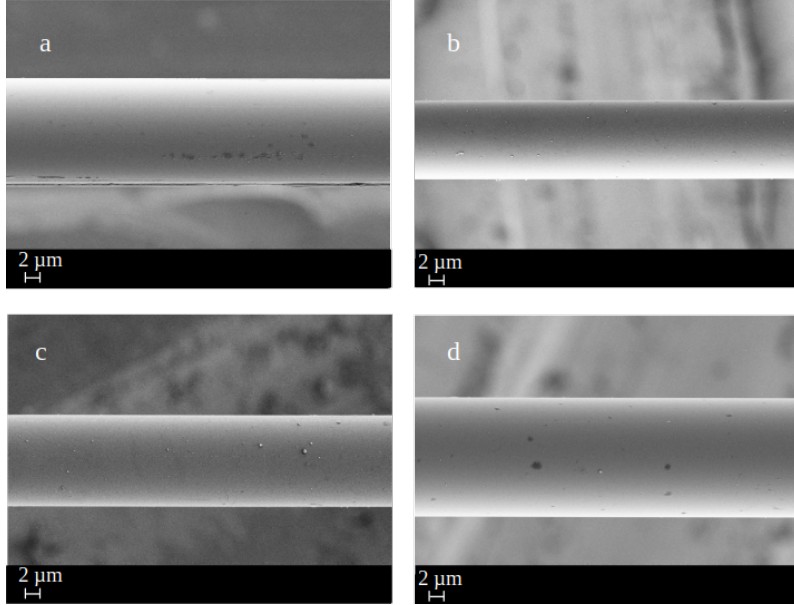

**Figure 7.** IPF fibers sized with AMEO (**a**) and coated with: SB (**b**), SB-MMT (**c**), and CSB (**d**). Magification 2.00 K×.

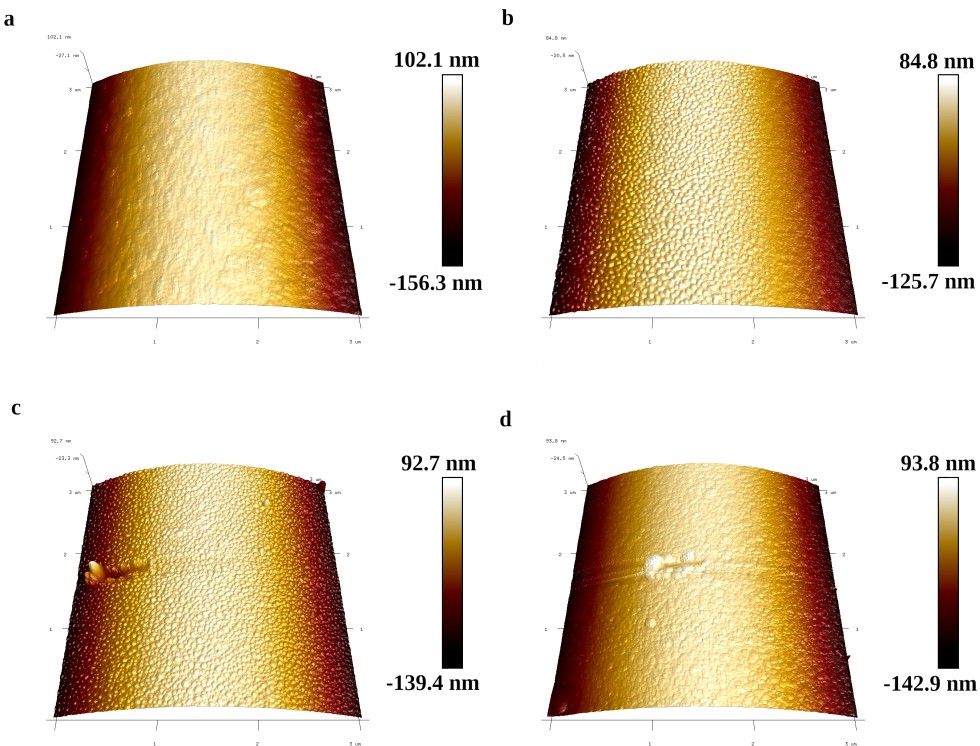

**Figure 8.** AFM images of IPF fibers coated with (**a**) SB, (**b**) CSB (**c**) SB-MMT, and (**d**) CSB-MMT.

**Table 2.** Average roughness $R_a$, square roughness $R_q$, and maximum roughness $R_{max}$ values of fibers with SB, SB-MMT, CSB, and CSB-MMT coating, determined by AFM.

| Fiber Type | $R_a$ [nm] | $R_q$ [nm] | $R_{max}$ [nm] |
|---|---|---|---|
| SB | 2.27 ± 0.29 | 3.33 ± 0.4 | 38.88 ± 32.34 |
| SB-MMT | 11.90 ± 1.17 | 16.91 ± 1.61 | 160 ± 19.65 |
| CSB | 2.46 ± 1.03 | 3.66 ± 1.91 | 46.05 ± 30.96 |
| CSB-MMT | 1.92 ± 0.5 | 3.70 ± 1.22 | 107.20 ± 26.93 |

The surface of AR-glass fibers coated with the self-crosslinked styrene–butadiene copolymer SB exhibits a quite smooth coating (Figure 8a), showing a rather homogeneous material distribution in accordance with Figure 7b. The same coating with 5 wt% MMT (Figure 8b) exhibits a marked increase of the surface roughness. A similar change in the surface morphology can be observed when the methylol melamin cross-linker is added to the carboxylated-styren butadiene coating (Figure 8c). It can be observed that, by adding both 5 wt% MMT and the methylol melamin cross-linking agent, no significant enhancement of the surface roughness is observed (Figure 8d). On the contrary, a pronounced increase in the surface roughness was observed by adding a 5 wt% of MMT in the SB polymer. Indeed, the highest average roughness, square roughness, and maximum roughness are obtained for fibers SB-MMT, and they are three to four times higher than those obtained for the same polymer without nanoclay particles. An increase in the maximum roughness is also observed for CSB-MMT. However, the square roughness is similar to that obtained for fibers CSB and the $R_a$ parameter is the lowest obtained. This confirms the presence of nanoclay agglomerate in the CSB-MMT coating, rather than a uniform distribution of the MMT particles in the polymeric matrix.

### 3.3. Single Fiber Tensile Strength

Table 3 reports the mechanical properties obtained from the single fiber tensile test of the IPF AR-glass fibers, unsized and after application of sizing and coating SB-MMT and CSB. It can be observed that the most significant increase in the tensile strength was obtained after application of the sizing. Compared to the unsized fibers, PTMO fibers have a 50% higher tensile strength. This is in accordance with results obtained in other studies and can be attributed to the ability of the sizing to increase the fiber tensile strength by mitigating the effect of existing surface defect and preventing the formation of new surface flaws [70].

**Table 3.** Mechanical properties of the IPF AR-glass fibers, unsized, sized with PTMO, and coated with either SB-MMT or CSB.

| Sample | Young's Modulus [MPa] | Tensile Strength [MPa] | Strain at Break [%] |
|---|---|---|---|
| unsized | 76.14 ± 2.5 | 1018.48 ± 379.23 | 1.51 ± 0.58 |
| PTMO | 76.16 ± 1.00 | 1534.47 ± 433.88 | 2.35 ± 0.71 |
| SB-MMT | 76.34 ± 0.85 | 1520.79 ± 392.86 | 2.29 ± 0.62 |
| CSB | 77.06 ± 0.89 | 1578.88 ± 371.82 | 2.35 ± 0.58 |

This phenomenon is usually called *healing effect* and depends on the ability of the sizing to penetrate in the surface defects. Therefore, the *healing effect* is correlated with the dimension of the sizing particles. Pure silane dispersions have been proved to enable healing a great part of the defect content [71], which is in accordance to the results obtained for unsized and PTMO fibers. The results obtained for SB-MMT and CSB suggest that the application of coatings on sized fibers does not lead to a further reduction of the number and severity of the surface flaws.

### 3.4. Formation of Hydration Products

Figure 9 shows the surface of roving and single fibers coated with SB and SB-MMT. The formation of crystals could be observed only on the surface of the fiber modified with the coating containing montmorillonite. This confirms the ability of nanoclay particles to act as nucleation points for the formation of hydration products, as reported in other studies [29,30].

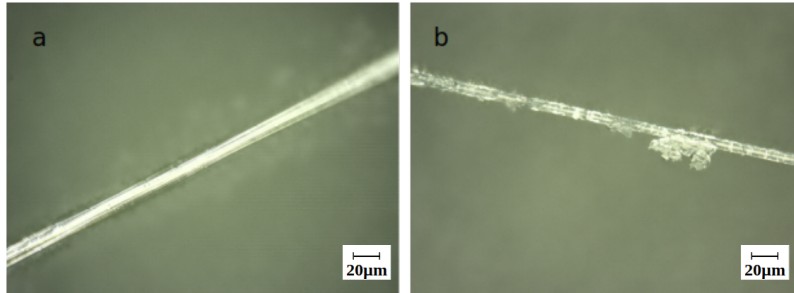

**Figure 9.** Microscope images of the glass SB fiber (**a**) and SB-MMT fiber (**b**) after immersion in cement solution overnight.

### 3.5. Fiber Matrix Interaction

During fiber pull-out the crack propagates along the embedded fiber with frictional slip in the debonded regions, as described in detail in Reference [72]. The first part of the force-displacement curve is characterized by linear elastic deformation, followed by the first crack during interfacial debonding that is usually revealed by a kink in the graph, especially when high-modulus fibers and low modulus polymer matrices are combined. As already discussed in previous works [58], typically, therefore, no "kink" does occur if

AR-glass fiber is pulled out of a cementitious matrix (Figure 10). After debonding, the applied force increases, as results of both interfacial adhesion and friction, until it reaches a peak value ($F_{max}$). At some point, the interfacial adhesion and friction are no longer in equilibrium, which leads to uncontrolled debonding until the frictional bond is reached, indicated by a characteristic force drop, which is particularly pronounced for high chemical fiber-matrix bonding. From this point to full fiber pull-out, the measured force value is determined by fiber-matrix friction that is here observed as the most dominant form of fiber-matrix interaction in the measured pull-out curves for all treated fibers. For this reason, the area under the graph that corresponds to the total pull-out work $W_{total}$ was considered to characterize the efficiency of the different surface treatments. Additionally, the interfacial shear strength (IFSS), defined by

$$IFFS = \frac{F_{max}}{\pi \cdot d_f \cdot l_e},$$ (5)

where $d_f$ is the fiber diameter, and $l_e$ the embedded fiber length, was determined to evaluate the fiber-matrix interaction; see Table 4.

For all the fibers, the maximum force was reached at low displacement levels due to the brittle nature of the fiber-matrix combination. The silane-treated fibers (PTMO) (Figure 10a) display the lowest $W_{total}$ but the highest $F_{max}$ value and IFSS. This means that the PTMO fibers have a stronger interfacial adhesion than the coated fibers. This can be attributed to the ability of the silane groups to build up chemical bonds to the OH-groups provided not only by the fiber surface, but also by the components of the cementitious matrix. By applying some coating on the fibers surface, the IFSS decreases, but, at the same time, a significant increase in the pull-out work can be observed for all the coated fibers. This can be attributed to the enhancement of the interfacial friction, which is particularly evident observing the shape of the second part of the graph. Here, the fiber is completely debonded and the measured force is the result of only frictional interaction between the fiber and the matrix. Comparing the pull-out graphs in Figure 10, it can be observed that, on average, after debonding, the coated fibers display higher pull-out load than PTMO fibers. During the pull-out, the coating is partially removed from the coating surface, and it accumulates at the fiber-matrix interface. Therefore, the greater friction between the matrix and the coated fibers can be considered the results of mechanical interlocking that arises during ongoing pull-out due to the removed coating material that remains between the fiber surface and the matrix. According to the results obtained for SB and CSB (Table 4), the addition of the methylol melamin cross-linking agent does not have significant effect on the pull-out behavior of the modified fibers. Therefore, it can be concluded that the increase in the fiber surface roughness observed in the AFM investigation (Table 3) plays a minor role in the quasi static pull-out for the fiber-matrix system investigated. On the contrary, the addition of nanoclay particles in both the SB and CSB polymers determine an increase of both the IFSS and $W_{total}$. This can be attributed to the ability of the MMT particles to accelerate and enhance the formation of hydration products on the fiber surface, as shown during the storage in cement solution in Section 3.4. The results confirm the hypothesis that nanoclay particles can increase the interaction between organic coating and cementitious matrix.

**Table 4.** Results of SFPO depending on the surface treatment of the ARG fibers.

| Sample Name | IFSS [MPa] | $W_{total}$ [mN*mm] | $l_e$ [mm] |
|---|---|---|---|
| PTMO | 4.2 ± 2.4 | 15.2 ± 7.6 | 663 ± 245 |
| SB | 1.9 ± 0.3 | 25.3 ± 5.6 | 894 ± 163 |
| CSB | 1.5 ± 0.5 | 20.3 ± 7.5 | 847 ± 106 |
| SB-MMT | 2.2 ± 0.5 | 28.8 ± 10.1 | 782 ± 117 |
| CSB-MMT | 1.9 ± 0.5 | 34.4 ± 11.3 | 883 ± 62 |

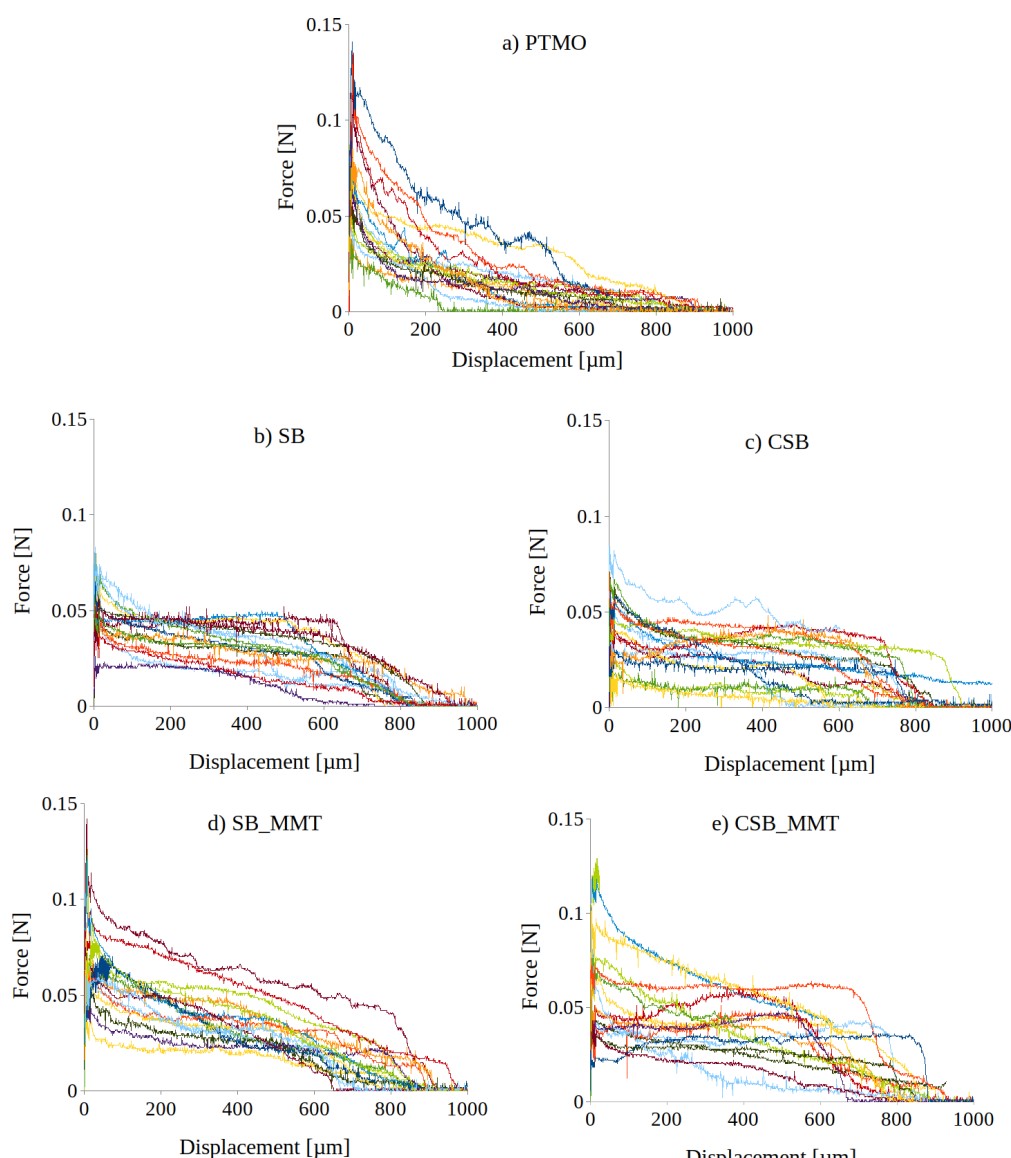

**Figure 10.** Force-displacement curves of single fiber pull-out tests, with the different colors of the curves relative to the individual samples tested.

## 4. Conclusions

This work provides new insight on the ability of nanoclay particles to improve the interfacial bond of coated fiber with a cementitious matrix. The fiber-matrix interaction was investigated by means of single fiber pull-out tests. The morphological and chemical investigations of the coatings were compared with the results obtained from the load displacement curves in order to obtain a deeper insight in the factors that influence the fiber pull-out behavior. From the results obtained, the following conclusions can be drawn:

- The dispersion quality of nanoclay particles in the coating can be affected by further ingredients, as it was observed in this work for the cross linking agent.
- The single fiber tensile strength is mainly depending on the silane treatment; no positive effect was arising from the nanoclay containing polymer coatings.
- The silane-treated fibers (PTMO) display higher IFSS but lower $W_{total}$. This indicates that the PTMO fibers have a high interfacial adhesion due to the chemical bonding with the matrix. However, the resulting force drop after reaching $F_{max}$ indicates only minor fiber-matrix interaction during further pull-out; hence, brittle failure occurs.

- The application of a coating increases the friction between the fiber and the matrix resulting in higher pull-out work.
- For both SB and CSB coatings, the addition of MMT particles produced an increase of the IFSS and the pull-out work. This can be attributed to a better interaction with the cementitious matrix and to the ability of nanoclay particles to work as nucleating species for the formation of hydration products.
- The increase of the polymer roughness did not lead to an increase of the pull-out load. This suggests that the modification of the chemical characteristics have a major impact on the interaction with the cementitious matrix.
- According to the results presented, the main benefit obtained from the dispersion of nanoclay particles in the coating consists in providing a better interaction based on chemical bonding in combination with improved mechanical interlocking. However, the potential, in terms of durability and mechanical performances, of engineering the fiber-matrix interface with a nanostructured coating are much greater. Further studies are certainly needed to better explore the applications of this technology. Particularly, future works should focus on the improvement of the coating in order to obtain a better dispersion of the nanoclay particles in the polymer. For this purpose, different protocols for the preparation of the coating can be considered. In addition, the possibility to use other polymers and nanoclay particles can be examined. Moreover, the optimal amount of nanoclay dispersed in the coating should be defined. Finally, the results obtained on a microscale should be confirmed by investigation conducted on a macroscale.

**Author Contributions:** Conceptualization, F.B. and C.S.; methodology, F.B. and C.S.; investigation, F.B.; resources, F.B.; data curation, F.B., C.S., and T.U.; writing—original draft preparation, F.B. and C.S.; writing—review and editing, C.S., T.U., and J.D.; supervision, C.S. and J.D.; project administration, C.S. All authors have read and agreed to the published version of the manuscript.

**Funding:** This research is partly funded by the German Research Foundation (Deutsche Forschungsgemeinschaft-DFG) within the framework of SFB/TRR280, Project-ID 417002380.

**Institutional Review Board Statement:** Not applicable.

**Informed Consent Statement:** Not applicable.

**Acknowledgments:** The authors greatly acknowledge the funding by the Deutsche Forschungsgemeinschaft (DFG, German Research Foundation) in the framework of the Collaborative Research Centre SFB-TRR280 "Design Strategies for Material-Minimized Carbon Reinforced Concrete Structures—Principles of a New Approach to Construction" project ID 417002380. The authors are grateful to Andreas Leuteritz for providing the nanoclay particles. The authors would also like to thank Regine Boldt, Steffi Preßler, Alma Rothe, Matthias Krüger, and Adriano Di Cristoforo for their support in sample preparation, performing single fiber pull-out tests, and other investigations reported in this paper.

**Conflicts of Interest:** The authors declare no conflict of interest.

## Abbreviations

The following abbreviations and symbols are used in this manuscript:

| | |
|---|---|
| FRCM | Fabric Reinforced Cementitious Matrix |
| TRC | Textile Reinforced Concrete |
| AR-Glass | Alkali Resistant Glass |
| AFM | Atomic Force Microscopy |
| SEM | Scanning Electron Microscope |
| SEM-EDX | Scanning Electron Microscope—Energy-Dispersive X-ray |
| IFSS | Interfacial Shear Strength |
| SFPO | Single Fiber Pull-out |
| MMT | Montmorillonite |
| XRD | X-ray Diffraction |

| IPF | Institut für Polymerforschung |
| SB | Styrene-Butadiene |
| RH | Relative Humidity |
| PLS | Polymer Layered Silicate |
| VAc/VeVa/E | Vinyl acetate/vinyl esther of versatic acid/ethylene |
| $\theta$ | Diffraction angle |
| $\lambda$ | wave length of the X-ray radiation |
| $d$ | layered silicate interlayer |
| $R_a$ | average roughness |
| $R_q$ | square roughness |
| $R_{max}$ | maximum roughness |
| $F_{max}$ | maximum pull-out force |
| $W_{total}$ | total pull-out work |
| $d_f$ | fiber diameter |
| $l_d$ | embedded fiber length |

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
