# Peer review of "Polymeric Coatings for AR-Glass Fibers in Cement-Based Matrices: Effect of Nanoclay on the Fiber-Matrix Interaction"

_applsci, doi:10.3390/app11125484_

Round 1

Reviewer 1 Report

  1. The current study investigates the effect of using polymeric coatings in AR glass textile used in cement based composites. The authors aim to understand the impact of adding nano fillers as a mean of improving chemical compatibility between the coating and the composite matrix. The authors report that the addition of nano clay in the coating improved the chemical bonding with the matrix
  2. The abstract needs significant improvement. The important part of it start in line 9 and afterwards. The first nine lines do not add much importance to the abstract and be better suited in the introduction.
  3. Please consider reviewing the abstract and highlight the novelty, major findings and conclusions.
  4. After line 120 the authors are encouraged to answer the following question: What is the research gap did you find from the previous researchers in your field? Mention it properly. It will improve the strength of the article.
  5. Does the authors have any mechanical data for any of the materials used in the study from the open literature? A table of these properties would be a good addition in the materials and methods section
  6. Some of the parameters used in the study in different sections in the material and methods part can be better presented if add in tables rather than in text.
  7. The authors should add a list of nomenclature at the end of the manuscript for all the symbols and Greek letters reported in this study
  8. Line 284 “defraction” is this word correct? Please check
  9. Also for line 284 “defraction peak that is shifted to lower degrees” please explain this further it is not clear what do you mean by that
  10. Line 286 “swallow water molecules” is swallow the right word to use here? Perhaps you mean absorb?
  11. Line 309 “In order to obtained” obtain? Please consider checking the English of the paper many mistakes in the text, “gave a quite different results” remove quiet it does not add any value to the sentence
  12. Extensive editing of English language and style required
  13.  Line 328-330 can the authors support this statement with a reference(s) from the open literature or mention what did previous studies report about this.
  14. Please consider combining figures 6 and 7
  15. Line 355-256 “that the crosslinking agent does not react with the car…” why this assumption was made?
  16. Please consider combining figures 11 and 12
  17. Please consider enlarging figures 11 and 12 they are not easy to read, the data is too dense
  18. The legends are missing in figures 11 and 12
  19. The results are well described but somewhat limited to comparing the experimental observation. The authors are encouraged to include more detailed discussion and critically discuss the observations from this investigation with existing literature.

Author Response

The current study investigates the effect of using polymeric coatings in AR glass textile used in cement based composites. The authors aim to understand the impact of adding nano fillers as a mean of improving chemical compatibility between the coating and the composite matrix. The authors report that the addition of nano clay in the coating improved the chemical bonding with the matrix

  1. The abstract needs significant improvement. The important part of it start in line 9 and afterwards. The first nine lines do not add much importance to the abstract and be better suited in the introduction.

The abstract has been significantly improved in order to better describe the purpose of this work.

  1. Please consider reviewing the abstract and highlight the novelty, major findings and conclusions.

The abstract has been modified giving more emphasis to the innovative aspect of the study.

  1. After line 120 the authors are encouraged to answer the following question: What is the research gap did you find from the previous researchers in your field? Mention it properly. It will improve the strength of the article.

The research gap fin in the previous research has been highlighted. Please see Line 117-122

  1. Does the authors have any mechanical data for any of the materials used in the study from the open literature? A table of these properties would be a good addition in the materials and methods section

Some information about the mechanical properties of the cementitious matrix has been added. No information about the mechanical properties of other materials used are available.

  1. Some of the parameters used in the study in different sections in the material and methods part can be better presented if add in tables rather than in text.

Thanks for your suggestion. The section materials and methods has been modified, in order to improve the clarity of the protocols employed. Particularly the parameters used in equations 1, 2 and 3 have been listed under the equation rather than described in the text. However, no table has been added.

  1. The authors should add a list of nomenclature at the end of the manuscript for all the symbols and Greek letters reported in this study

A list with all symbols and Greek letters has been added at the end of the manuscript.

  1. Line 284 “defraction” is this word correct? Please check

    The word has been corrected.

  2. Also for line 284 “defraction peak that is shifted to lower degrees” please explain this further it is not clear what do you mean by that 

    The concept has been explained more in detail. Line 295-297

  3. Line 286 “swallow water molecules” is swallow the right word to use here? Perhaps you mean absorb? 

    Thanks for your suggestion, the term has been modified.

  4. Line 309 “In order to obtained” obtain? Please consider checking the English of the paper many mistakes in the text, “gave a quite different results” remove quiet it does not add any value to the sentence

Improper terms and spelling mistakes have been corrected.

  1. Extensive editing of English language and style required

An extensive English review as been done.

  1.  Line 328-330 can the authors support this statement with a reference(s) from the open literature or mention what did previous studies report about this.

Some references to other studies have been added.

  1. Please consider combining figures 6 and 7

It is the author opinion that figures 6 and 7 should not be combined since they can not be directly compared: figure 6 presents the elemental distribution of Carbon (a) and Nitrogen (b) in CSB_MM film, while figure 7 displays the SEM images of two fibers coated with CSB_MMT. However, we considered to combined figure 5 and 6.

  1. Line 355-256 “that the crosslinking agent does not react with the car…” why this assumption was made?

Some more detailed explanation have been added in the previous section. Line 343-353

  1. Please consider combining figures 11 and 12 Figures have been combined together.

  2. Please consider enlarging figures 11 and 12 they are not easy to read, the data is too dense

Figures have been enlarged.

  1. The legends are missing in figures 11 and 12

It is the authors opinion that in Figures 11 and 12 legends are not needed, since all the curves in the graph correspond to the same kind of samples. To make this more clear we added an explaining sentence. 

However, the figures have been enlarged to make the curves clearer and more visible.

  1. The results are well described but somewhat limited to comparing the experimental observation. The authors are encouraged to include more detailed discussion and critically discuss the observations from this investigation with existing literature.

More detailed discussion in reference to the existing literature have been included. Please see Lines: 342-344, 349-357.

Reviewer 2 Report

The mansucript ,,Polymeric coatings for AR-glass fibers in cement-based matrices: effect of nanoclay on the fiber-matrix interaction,, has interesting findings. However, the manuscript needs to be improved before publication. I recommend a major revision.

My advice, recommendations and comments are listed below.

Please be sure that your manuscript thoroughly establishes how this work is fundamentally novel. Specific comparisons should be made to previously published materials that have a similar purpose. Please present a strong case for how this work is a major advance. This needs to be done in the manuscript itself, not just in the response to review comments. This is a very important point in terms of which I will further consider the manuscript.

Please be sure that your abstract and your Conclusions section not only summarize the key findings of your work but also explain the specific ways in which this work fundamentally advances the field relative to prior literature.

The significance of this study should be more emphasize in the introduction.

See this documents, which may be helpful. https://journals.sagepub.com/doi/10.1177/1528083704039833

https://www.sciencedirect.com/science/article/pii/S1359645404003647

Line 26-29, copolymer: This important document has been addressed in this issue in detail, so authors are encouraged to add it here as a reference and improve the introduction. https://www.mdpi.com/2073-4360/12/3/708

Line 121: Indicate in the introduction whether it is a sodium or calcium form of MMT.

Line 138: Country of origin Germany, please specify.

Figure 2: Very well executed illustrative image.

Table 1: Are you mentioning the surface treatment, have you not even considered the intercalation method?

Line 181: ,, basal spacing,, please write in brackets (d001 value)

Line 200: Improve the notation of equations with explanations throughout the manuscript.

Line 213: Why did you make XRD recordings between 2 ° to 30 °  2 theta. What a  higher angles and important diffractions?

Figure 3: What do you attribute noise to in XRD recordings?

Line 298-302, intercalation of alkylammonium ions: This statement is supported by this very important document. Authors should add a reference that confirms this. https://www.sciencedirect.com/science/article/pii/S0169131719301413

Figure 5: If it is possible, improve quality of Figure 5.

Line 327: Have you performed a carbon analysis to determine the amount of these elements to the ratio of hydrated inorganic cations?

Figure 11 and 12: It is very difficult to read anything from this figure. Improve the quality and presentation of this figure.

Line 438: Indicate the possible risks of such research. Add your recommendations for future research.

References: Make sure the references are added correctly according to the journal's instructions.

Author Response

  1. Please be sure that your manuscript thoroughly establishes how this work is fundamentally novel. Specific comparisons should be made to previously published materials that have a similar purpose. Please present a strong case for how this work is a major advance. This needs to be done in the manuscript itself, not just in the response to review comments. This is a very important point in terms of which I will further consider the manuscript.

    The research need and novelity is now derived in lines 120-125.

  2. Please be sure that your abstract and your Conclusions section not only summarize the key findings of your work but also explain the specific ways in which this work fundamentally advances the field relative to prior literature.

    The abstract has been rewritten.

  3. The significance of this study should be more emphasize in the introduction. See this documents, which may be helpful. https://journals.sagepub.com/doi/10.1177/1528083704039833

https://www.sciencedirect.com/science/article/pii/S1359645404003647

The significance of the study was more emphasized in the abstract in the introduction and in the conclusions.

  1. Line 26-29, copolymer: This important document has been addressed in this issue in detail, so authors are encouraged to add it here as a reference and improve the introduction. https://www.mdpi.com/2073-4360/12/3/708

We really much appreciated the article you referenced, however in the authors’ opinion, the results reported in the study have no significant correlation with concepts reported in the introduction at Line 26-29.

Indeed in Line 26-29 we report that different studies and commercial solutions propose the employment of fabric impregnated with polymeric coatings in order to enhance the load bearing capacity of the reinforcement and protect the fibers from abrasion and chemical corrosion.

The article you reported described the synthesis of a low molecular weight zwitterionic copolymer for improving wellbore stability.

  1. Line 121: Indicate in the introduction whether it is a sodium or calcium form of MMT.

The information has been added in the introduction

  1. Line 138: Country of origin Germany, please specify.

The origin has been specified.

  1. Figure 2: Very well executed illustrative image.

    Thanks for your appreciation.

  2. Table 1: Are you mentioning the surface treatment, have you not even considered the intercalation method?

Table 1 describes the sizing/coating applied on the fibers surface. The intercalation method is illustrated in figure 2 and described in the text.

  1. Line 181: ,, basal spacing,, please write in brackets (d001 value)

Done.

  1. Line 200: Improve the notation of equations with explanations throughout the manuscript.

The equations have been cross-referenced in the text.

  1. Line 213: Why did you make XRD recordings between 2 ° to 30 °  2 theta. What a  higher angles and important diffractions?

XRD were extended to higher angles in order to exclude the presense of other phase and impurities. However since the diffraction peaks relevant for the discussion are in the range between 2-10° 2theta, the spectrum reported was limited to smaller angles.

  1. Figure 3: What do you attribute noise to in XRD recordings?

The noise in the XRD recordings is probably due to the amorphous structure of the polymer.

  1. Line 298-302, intercalation of alkylammonium ions: This statement is supported by this very important document. Authors should add a reference that confirms this. https://www.sciencedirect.com/science/article/pii/S0169131719301413

Thank you for your suggestion, the reference was added.

  1. Figure 5: If it is possible, improve quality of Figure 5.

Figure 5 has been modified.

  1. Line 327: Have you performed a carbon analysis to determine the amount of these elements to the ratio of hydrated inorganic cations?

The observation made are only based on the distribution of C N and Si in the images reported in the paper. The ratio between hydrated inorganic cations and carbon has not been determined.

  1. Figure 11 and 12: It is very difficult to read anything from this figure. Improve the quality and presentation of this figure.

The figures have been modified.

  1. Line 438: Indicate the possible risks of such research. Add your recommendations for future research.

Recommendation for future research has been added in the conclusions. Line 497-503

  1. References: Make sure the references are added correctly according to the journal's instructions.

The references have been checked.

Round 2

Reviewer 1 Report

The authors have answered all addressed questions raised in previous review;

just one more issue to check please:

The square roughness is usually written as Sq and not as Rq, could you please check this and revise if needed.

The manuscript still needs moderate English Editing.

Best wishes 

Author Response

Thanks for your revision.

Please find below the response to your comments.

The square roughness is usually written as Sq and not as Rq, could you please check this and revise if needed.

They are both used. See for example:

Tiantian Li, Chongyang Shen, Sen Wu, Chao Jin, Scott A. Bradford,
Synergies of surface roughness and hydration on colloid detachment in saturated porous media: Column and atomic force microscopy studies,
Water Research, Volume 183, 2020, 116068.

R.R.L. De Oliveira, D.A.C. Albuquerque, T.G.S. Cruz, F.M. Yamaji and F.L. Leite (2012). Measurement of the Nanoscale Roughness by Atomic Force Microscopy: Basic Principles and Applications, Atomic Force Microscopy - Imaging, Measuring and Manipulating Surfaces at the Atomic Scale, Dr. Victor Bellitto (Ed.)

The manuscript still needs moderate English Editing.

Further English corrections have been done. Please fine attached the last version of the paper.

Best wishes

Reviewer 2 Report

The manuscript ,,Polymeric coatings for AR-glass fibers in cement-based matrices: effect of nanoclay on the fiber-matrix interaction,, has been significantly improved. I recommend accepting the manuscript in its current form.

Author Response

Thanks for your revision.

Best wishes